# The effects of spectral dimensionality reduction on hyperspectral pixel classification: A case study

**Kiran Mantripragada**[1]*, **Phuong D. Dao**[2,3], **Yuhong He**[2], **Faisal Z. Qureshi**[1]

**1** Faculty of Science, University of Ontario Institute of Technology, Oshawa, ON, Canada, **2** Department of Geography, Geomatics and Environment, University of Toronto, Mississauga, ON, Canada, **3** School of the Environment, University of Toronto, Toronto, ON, Canada

* kiran.mantripragada@ontariotechu.net

**Data Availability Statement:** All Hyperspectral image files are available from the Visual Computing Lab - Ontario Tech University URL: http://vclab.science.uoit.ca/datasets/ut-hsi301/.

## Abstract

This paper presents a systematic study of the effects of hyperspectral pixel dimensionality reduction on the pixel classification task. We use five dimensionality reduction methods—PCA, KPCA, ICA, AE, and DAE—to compress 301-dimensional hyperspectral pixels. Compressed pixels are subsequently used to perform pixel classifications. Pixel classification accuracies together with compression method, compression rates, and reconstruction errors provide a new lens to study the suitability of a compression method for the task of pixel classification. We use three high-resolution hyperspectral image datasets, representing three common landscape types (i.e. urban, transitional suburban, and forests) collected by the Remote Sensing and Spatial Ecosystem Modeling laboratory of the University of Toronto. We found that PCA, KPCA, and ICA post greater signal reconstruction capability; however, when compression rates are more than 90% these methods show lower classification scores. AE and DAE methods post better classification accuracy at 95% compression rate, however their performance drops as compression rate approaches 97%. Our results suggest that both the compression method and the compression rate are important considerations when designing a hyperspectral pixel classification pipeline.

## 1 Introduction

There is a growing demand for improved hyperspectral image analysis in part due to increasing availability of images with high spatial and spectral resolutions [1, 2]. Hyperspectral images capture information from the ultraviolet, visible, and infrared regions of the electromagnetic waves and record the spectral signature of the observed objects. The richness of information in the spectra is helpful in a variety of tasks such as object detection, segmentation, and classification. Consequently, hyperspectral images have found wide-spread use in a number of application domains, such as mining, environmental monitoring, military, etc. Unlike ordinary images, in which each pixel consists of 3 channels (red, green, and blue), a pixel in a hyperspectral image can consist of upwards of 300 spectral data values. This suggests that hyperspectral images have much higher requirements in terms of storage space and computational

**Funding:** Natural Sciences and Engineering Research Council of Canada (NSERC) through the NSERC Discovery Program: Funding of the hyperspectral Image acquisition mission and image preprocessing facility (RGPIN-386183 awarded to Dr. Yuhong He) - Visual Computing Lab of the Ontario Tech University (RGPIN-2020-05159, awarded to Dr. Faisal Z. Qureshi). The funders had no role in study design, data collection and analysis, decision to publish, or preparation of the manuscript.

**Competing interests:** The authors have declared that no competing interests.

processing. Therefore, the search for better methods for hyperspectral image storage, processing, and analysis continues unabated.

Spectral information stored at each pixel is often redundant [3], therefore it is often not necessary to process all spectral bands when performing hyperspectral image segmentation or classification. This is especially true when hyperspectral images represent regions that contain a particular set of materials, often referred to as *endmembers* [4]. A first step towards developing computationally efficient techniques for hyperspectral image processing is to reduce the inherent redundancy in hyperspectral images, thereby reducing the amount of data that needs to be processed in the subsequent steps. Within this context, feature extraction, band selection, and compression have grown into active research areas. A number of researchers have explored dimensionality reduction techniques, such as Principal Component Analysis (PCA), Kernel PCA (KPCA), and Independent Component Analysis (ICA), to compress hyperspectral image pixels with a view to reducing the redundancy inherent in these images [4–6]. Autoencoder models have also been used to construct low-dimensional features that are subsequently used for image analysis tasks [7–9].

The compression techniques used in hyperspectral image analysis aim to find the lower-dimensional encoding of the spectral signal by minimizing the reconstruction loss. This is a logical choice since it ensures that the low-dimensional encoding retains the important information needed to reconstruct the original spectral signal with minimal distortion or loss. We instead argue that a better approach to selecting the best compression method is to study the performance of the penultimate task—in our case, pixel classification—on the compressed signal. Specifically, we seek a compression method that encodes the spectral signal in a low-dimensional space such that the low-dimensional encoding both minimizes the reconstruction loss and maximizes the pixel classification performance. This idea is motivated by lossy image compression approaches, e.g., the Joint Photographic Experts Group (JPEG) standard, that balance perceptual loss against compression rates. Note that this work is concerned with pixel level compression. Image level compression should also exploit spatial information encoded within neighbouring pixels, and this work does not consider spatial information.

We setup the problem as follows. First, we assert that the performance of the final task that we want to carry out is a better proxy for evaluating the performance of an image compression algorithm. A central objective of many hyperspectral image analysis methods is to achieve better pixel-level classification; therefore, we decided to use pixel classification to study the performance of five widely used compression techniques for hyperspectral pixels: three dimensionality reduction methods—PCA, KPCA, and ICA–and two deep learning based approaches—Autoencoder (AE) and denoising autoencoder (DAE)—construct low-dimensional encodings of the input pixel at various compression rates ranging from 1% to 99%. These encodings are subsequently used for pixel label classification. To the best of our knowledge, this is the first systematic study that captures the interplay between compression methods and rates and the task of hyperspectral pixel classification. We use three new hyperspectal images, each representing a common landscape type (i.e. urban, transitional suburban, and forests) collected by the Remote Sensing and Spatial Ecosystem Modeling laboratory of the University of Toronto, to carry out the experiments (see Section 3). Those who are interested in computationally efficient hyperspectral pixel analysis will find our findings useful.

The paper makes the following contributions: it presents a first-of-its-kind study of the effects of hyperspectral pixel compression on pixel classification. The paper studies classification performance when hyperspectral pixels are compressed using one of five compression methods using various rates of compression. The results suggest that AE and DAE methods create pixel encodings that achieve best classification scores for compression rates around

95%. Additionally, we find that widely-used denoising filters are not needed when using AE or DAE methods for pixel compression.

The rest of this paper is organized as follows. The next section briefly summarizes prior work. Section 3 describes the three datasets that we used to carry out the experiments. Compression and classification methods are presented in Section 4, followed by results in Section 5 and conclusions in Section 6.

## 2 Background

Data compression and pixel level classification are important topics in hyperspectral image analysis. Pixel level classification is often called *semantic segmentation* in the wider computer vision literature. Hyperspectral images store two orders of magnitude more information than an ordinary RGB image, and it is desirable to compress these images to reduce storage requirements, improve processing speeds, and lower computational requirements. In addition it is sometimes possible to achieve satisfactory pixel classification performance even when using a fraction of spectral information available for a pixel [3, 10, 11]. Spectral compression is often the first step in hyperspectral image classification pipeline [12].

Dimensionality reduction algorithms PCA and ICA are widely used in the hyperspectral image analysis community for the purposes of reducing the number of channels per pixel prior subsequent analysis steps, such as image segmentation and pixel classification [1, 12–17]. Others have employed non-linear compression techniques, such as ICA [18] and Wavelet transform [19, 20], for compressing hyperspectral images. Dua *et. al.* provides a survey of various compression methods for hyperspectral images [21]. Local Linear Embedding (LLE) [22, 23], Laplacian Eigenmaps [23], image quantization techniques [21, 24], and compressing a sequence of hyperspectral images together (sometimes referred to as *temporal compression*) [21] have also been used in the hyperspectral image analysis community to reduce the amount of data that needs to be stored and processed.

A common class of methods for "compressing" hyperspectral images is *band selection* [25]. Band selection methods are used for a variety of analysis tasks in hyperspectral images, including ranking, searching, clustering, constructing sparse representations, etc. Farrell and Mersereau studied the impact of PCA on hyperspectral images classification when targets pixels have similar spectral profile to those of background pixels. They found that the PCA compression had negligible effect on the performance of various classification methods. This work; however, did not study the classification performance as compression rate is varied.

As stated earlier, PCA is a commonly used compression method for hyperspectral images. PCA is a linear method, whereas it is well-known that the relationship between various "bands" of a spectral is highly non-linear. There are many reasons for it, including reflection, refraction, and the absorption property of materials that are being imaged, plus the noise inherent in the system due to atmospheric absorption and scattering. Cheriyadat and Bruce demonstrated the negative effects of PCA when used as pre-processing step for classification tasks [26]. Du *et al.* used ICA as a compression step for 6-band hyperspectral image classification [4]. Here, the compressed image comprised 4 bands. The authors noted that classification scores when using ICA with manual band-selection were better than the classification scores obtained when using PCA. They also showed that the classification performance using ICA with manual band-selection was worse than the classification performance on the full 6-band image.

More recently, the use of autoencoder methods to compress HSI is also increasing. Ball *et al.* mentioned the use of AEs for dimensionality reduction or for Remote Sensing datasets and not only HSI, while Paoletti *et al.* [27] described the rise of AE-based compression as a

critical pre-processing step for HSI Hyperspectral pixels. Zhang *et al.* proposed an AE model for compression in a pipeline for unsupervised learning. However, the previous authors usually selected the best compression rate for their tasks and did not evaluate the variation on the results to the entire range of compression rates.

## 3 Hyperspectral datasets

We used three high spatial resolution hyperspectral images for the studies presented in this paper (Fig 1). These images and annotations were used for the first time by Dao *et al.* [28], were captured using the Micro-HyperSpec III sensor (from Headwall Photonics Inc., USA) mounted at the bottom of a helicopter. The images were captured during the daytime at 10:30 am on August 20, 2017. The original images with 325 bands were resampled to obtain 301 bands from 400 nm to 1000 nm with an interval of 2 nm. Raw images were converted to at-sensor radiance using HyperSpec III software. The images were also atmospherically corrected to surface reflectance using the empirical line calibration method [29] with field spectral reflectance measured by FieldSpec 3 spectroradiometer from Malvern Panalytical, Malvern, United Kingdom. These images represent 1) urban, 2) transitional suburban, and 3) forests landcover types. These three landcover types cover a large fraction of use cases for hyperspectral imagery; urban and sub-urban images are often used for city planning and land use analysis and forest images are typically used for forest management, ecological monitoring, and vegetation analysis. The overlaid polygons in Fig 1 depict the annotated regions for which ground-truth pixel labels are available.

Fig 1 (second row, left) shows the hyperspectral image collected in an urban-rural transitional area. We refer to this image as "Suburban" dataset. It was captured around Bolton area in southern Ontario and covers an area between 43°52′32″ and 43°53′04″ in latitude and −79° 44′15″ and −79°43′34″ in longitude. This region consists of various land cover types, such as rooftops, asphalt roads, swimming pools, ponds, grassland, shrubs, urban forest, etc. The image also contains regions that are in shadows. The image resolution is 0.3 square meters and the covered area is around 41, 182 square meters. Table 1 shows the number of samples (pixels) for different landcover types used for training and testing.

Fig 1 (second row, middle) shows the hyperspectral image collected in a residential urban area, also around Bolton region in southern Ontario. We refer to this image as "Urban" dataset. It contains rooftops, under-construction residences, roads, and lawns landcover types. The dataset also exhibits regions that are in shadows. This image covers the area between 43°45′30″ and 43°45′43″ in latitude and −79°50′06″ and −79°49′51″ in longitude. The image resolution is 0.3 square meters and the area after removing background pixels is around 59, 834 square meters. Table 2 shows the number of samples (pixels) for different landcover types used for training and testing.

Fig 1 (second row, right) shows the hyperspectral dataset collected in a natural forest located at a biological site of the University of Toronto in King City region in southern Ontario. We refer to this dataset as "Forest" dataset. It covers the area between 44°01′58″ and 44°02′04″ in latitude and −79°32′06″ and −79°31′55″ in longitude. The image resolution is 0.3 square meters and the area after removing background pixels is around 43, 084 square meters. Table 3 shows the number of samples (pixels) for different landcover types used for training and testing.

## 4 Methodology

We used the following five methods to compress pixel spectral signal: 1) PCA, 2) KPCA, 3) ICA, 4) AE, and 5) DAE. We also trained a gradient boosted tree model to classify the

(A)

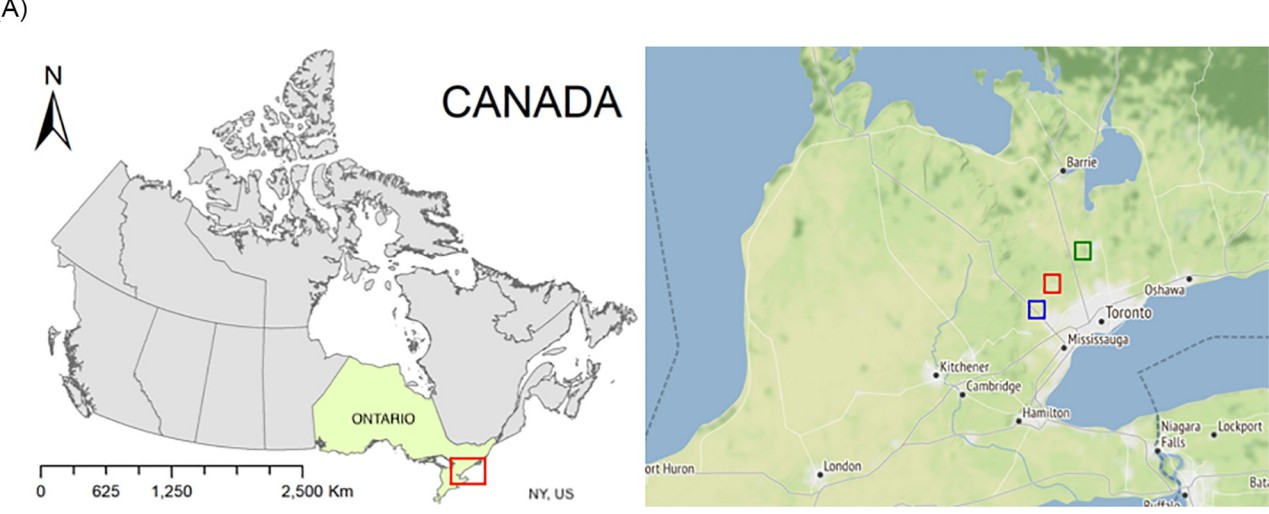

(B)

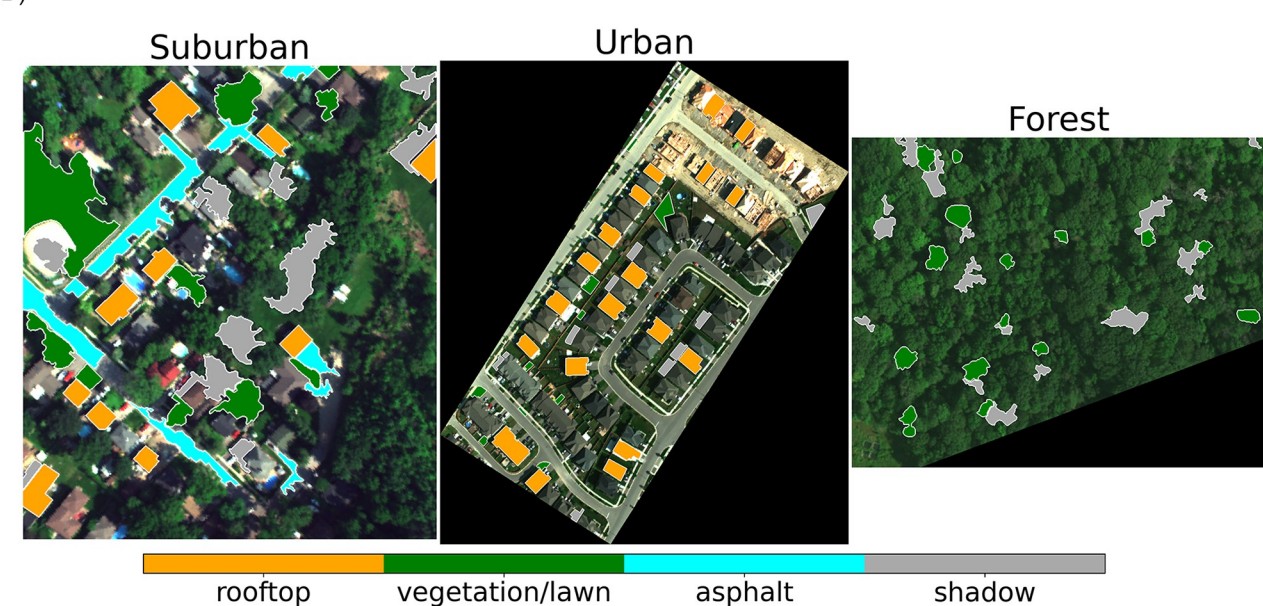

**Fig 1. Hyperspectral datasets were collected by the Remote Sensing and Spatial Ecosystem Modeling (RSSEM) Laboratory Department of Geography, Geomatics and Environment, University of Toronto using an airborne sensor over an area around Toronto, Ontario, Canada. Top left**: Study area; **Top right**: the red, blue, and green areas represent Suburban, Urban, and Forest images, respectively (the rectangles are not to scale). **Bottom row**: shows the three datasets in pseudo color (RGB images). This visualization was constructed using the 670 nm (red), 540 nm (green), and 470 nm (blue) bands from original HSI data. The yellow, green, blue, and gray polygons overlaid on the hyperspectral images are the areas for which ground-truth pixel labels available.

hyperspectral image pixels given their compressed signal. In addition, we measured the reconstruction errors by recovering the original pixel spectra from its compressed signal. Mathematically, $\mathbf{x}_i$ represents the hyperspectral pixel $i$. Here, $\mathbf{x}_i \in \mathbb{R}^D$ is a $D$-dimensional vector of real numbers. The three datasets used in this paper, namely Suburban, Urban, and Forest (Section 3), have dimensionality $D = 301$, $D = 301$, and $D = 251$, respectively.

**Table 1. Splits of train, test, and validation samples for Suburban dataset.**

| label | train | validation | test |
|---|---|---|---|
| Asphalt | 9155 | 4578 | 4578 |
| Rooftop | 7910 | 3955 | 3955 |
| Shadow | 10385 | 5192 | 5193 |
| Vegetation | 15147 | 7573 | 7574 |

**Table 2. Splits of train, test, and validation samples for Urban dataset.**

| label | train | validation | test |
|---|---|---|---|
| Lawn | 3432 | 1716 | 1716 |
| Rooftop | 22323 | 11162 | 11162 |
| Shadow | 4384 | 2192 | 2192 |

We used a compression method $\mathcal{E}$ to construct the compressed signal $\mathbf{z}_i = \mathcal{E}(\mathbf{x}_i)$, where $\mathbf{z}_i \in \mathbb{R}^d$ and $\mathcal{E}$ is one of the following: PCA, KPCA, ICA, AE, or DAE. Here $1 \leq d < 301$ is a controllable parameter and lower values of $d$ means higher compression rates. We computed classification labels $\mathcal{C}(\mathbf{z}_i)$ for pixel $i$ using its compressed signal, where $\mathcal{C}$ is the gradient boosted tree classifier. We were able to recover the original signal $\hat{\mathbf{x}}_i$ from $\mathbf{z}_i$ and computed the reconstruction error as $\|\hat{\mathbf{x}}_i - \mathbf{x}_i\|^2$, *i.e.* the Euclidean distance between the original pixel $\mathbf{x}_i$ and the reconstructed pixel $\hat{\mathbf{x}}_i$.

## 4.1 Compression methods

Below, we discuss the compression methods used in this paper–PCA, KPCA, and ICA– which have been widely used as dimensionality reduction methods. The two autoencoder models (AE and DAE) used in this study are discussed later in the section.

**4.1.1 PCA.** PCA projects the data onto a feature space that consists of the eigenvectors of the data covariance matrix. Dimensionality reduction is achieved by discarding data dimensions with low variance. The intuition being that data dimensions that exhibit low variance contains little useful information. We refer the reader to [30, 31] for more information on PCA.

**4.1.2 KPCA.** Kernel PCA is an extension of the PCA. Here, input data are mapped to a higher dimensional space using a kernel. As per *Vapnik-Chervonenkis* theory, data mapped to a higher dimensional space provide better separability. Popular kernel choices are Gaussian, Polynomial, Radial Basis Functions, and Hyperbolic Tangent. In this work, we used a polynomial kernel, which is well-suited to capture any non-linearities present in the data. More information about KPCA is available in [5, 32–34].

**4.1.3 ICA.** ICA decomposes the input signal into additive subcomponents under the non-Gaussian and statistical independence assumptions. It is then possible to represent the original signal using a subset of the independent components returned by the ICA method, thereby performing data compression. We refer the reader to [4, 35–38] for further details on ICA.

**Table 3. Splits of train, test, and validation samples for Forest dataset.**

| label | train | validation | test |
|---|---|---|---|
| Shadow | 9200 | 4600 | 4600 |
| Tree | 7343 | 3672 | 3672 |

**4.1.4 AE.** We used the AE model proposed by Hinton *et al.* [39]. It consists of two parts: 1) an encoder, which transforms the input signal **x** into a lower-dimensional signal **z**; and 2) a decoder, which reconstructs the original signal $\hat{\mathbf{x}}$ from the latent representation **z**. Specifically, the encoder contains of a single hidden layer, and it transforms 301 dimensional pixel spectra into a $d$ dimensional vector. The decoder also consists of a single hidden layer, and it reconstructs the 301 dimensional signal from a $d$ dimensional vector. Both encoder and decoder use ReLU (Rectified Linear Unit) activation functions for the hidden layers. The output layer of the decoder uses the Sigmoid activation function as the expected values of the reconstructed signal are restricted to the values of reflectance, i.e., between 0 and 1. We refer the reader to [40] for technical details about our autoencoder model. The number of elements (i.e., neurons) in the hidden layer is a hyperparameter. We used the grid search approach to estimate a "good" value for this hyperparameter. During hyperparameter selection we set the compression rate equal to 99% (i.e., $d$ was set to 4).

**4.1.5 Denoising AE.** It is well-known that hyperspectral images exhibit a higher degree of noise as compared to the noise present in ordinary RGB images. Furthermore, the level of noise present in different bands of a hyperspectral image varies between bands. Atmospheric water vapor, for example, affects near-infrared bands more than higher frequency bands. If left untreated, noise will place an adverse effect on the subsequent processing and analysis tasks, such as compression, segmentation, or classification. We implemented a denoising autoencoder, which accounts for the noise present in the signal, for compressing the input spectral signal [9, 41]. Denoising autoencoder also consists of an encoder and a decoder. The encoder consists of two hidden layers. The first hidden layer contains 400 neurons and the second hidden layer contains 500 neurons. The decoder also consists of two hidden layers. The first hidden layer contains 500 and the second hidden layer contains 400 neurons. All hidden layers use ReLU activation function. Decoder's output layer uses Sigmoid activation function.

In order to understand the effect of noise on hyperspectral images, we selected Saviszky-Golay (SG) algorithm, a widely used noise filtering method for hyperspectral images, to construct clean spectral signals [42, 43]. Fig 2 shows spectral curves for a randomly selected pixel in the three datasets. Fig 2 (middle) suggests that HSI+SG+DAE model did poorly in signal reconstruction, especially in the 500–800 nm range. We observe a similar trend for other pixels in the dataset. It appears that SG+DAE strongly attenuates the signal in this range. This confirms that it is unnecessary and perhaps counter-productive to use a denoising preprocessing step when using a denoising autoencoder to compress hyperspectral signal. The second row of Fig 2 shows the SNR (Signal-to-Noise ratio) of each compression method compared to the original HSI image. This result demonstrates that AE and DAE methods can improve the SNR of the signal, suggesting that a pre-processing step for denoising is not necessary when AE or DAE is used as a compression algorithms.

**4.1.6 Training regime for AE and DAE.** Both autoencoder and denoising autoencoder were trained using reconstruction loss, which is defined as $\|\hat{\mathbf{x}}_i - \mathbf{x}_i\|^2$. In our experiments, both autoencoders were able to achieve low reconstruction errors even for high compression rates. Tables 1–3 list the number of training and testing samples for the suburban, urban, and forest datasets, respectively. Each model was trained for 30 epochs using Adam optimizer. We trained each model ten times to capture the model variance. Fig 3 shows reconstruction errors for the three datasets for ten different runs for AE and DAE models. As expected, the reconstruction errors for DAE models exhibit a larger variance than those for AE models. For each image we selected the model with the lowest reconstruction error to be used as the compression method in the final classification pipeline.

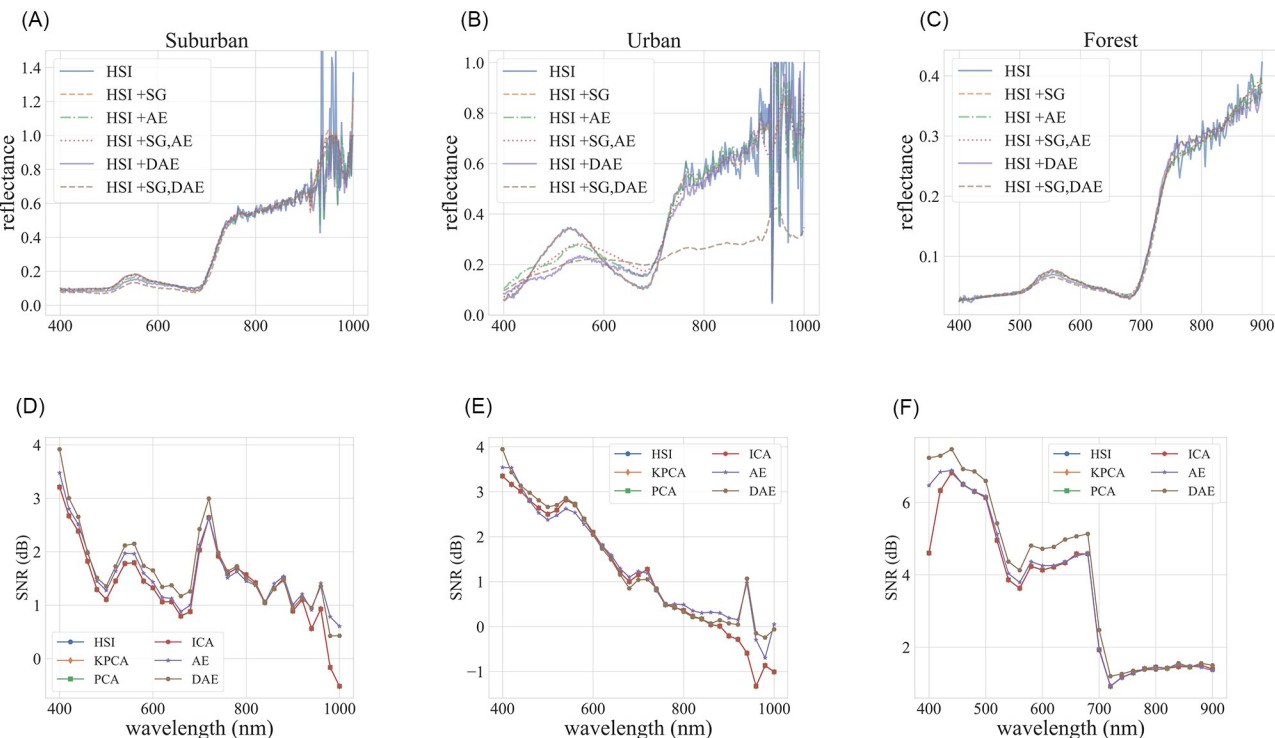

**Fig 2. First row**: Spectral reconstructions for a randomly selected pixel in the three images. HSI denotes the original spectral signal. HSI+SG refers to the denoised spectral signal. HSI+AE and HSI+DAE denote reconstructed spectral signals using autoencoder and denoising autoencoder, respectively. HSI+SG+AE and HSI+SG+DAE denote reconstructed spectral signal using transformed encodings (0% Compression rates) from denoised signals (HSI+SG). **Second row**: Signal-to-Noise Ratio of the reconstructed spectra (PCA, KPCA, ICA, AE, DAE) compared to the original pixel (HSI).

## 4.2 Gradient Boosted Tree Classifier

We employed a Gradient Boosted Tree (XGBClassifier) classifier for pixel classification [44, 45]. XGBClassifier is a widely used ensemble model and similar to other ensemble methods, it

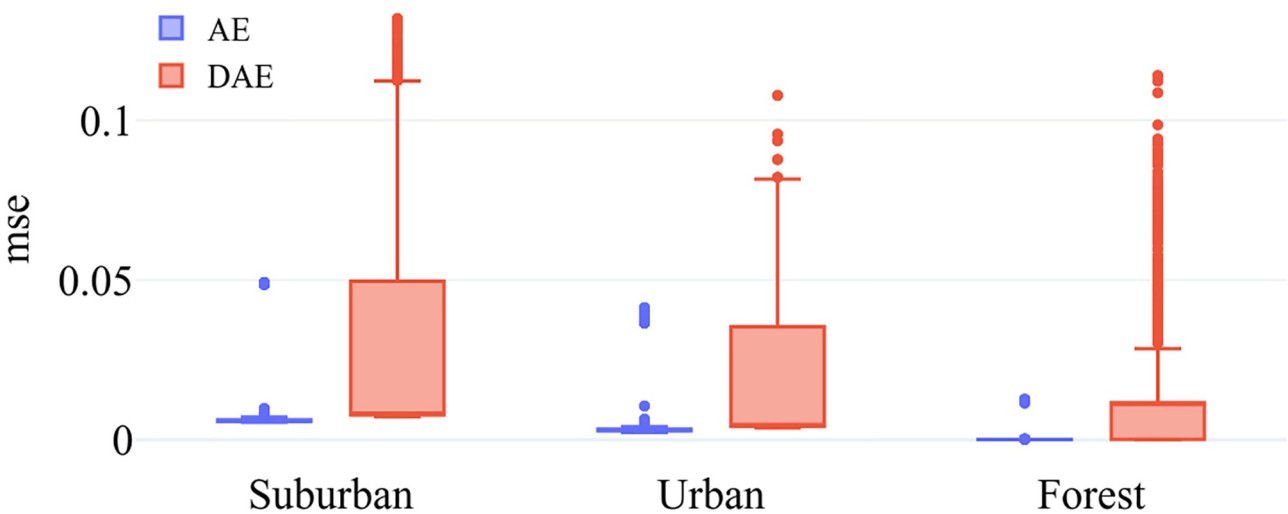

**Fig 3. Model variance.** Reconstruction errors for AE and DAE models for ten training runs.

avoids overfitting and offers good generalization properties [46]. It is also easy to construct intuitive interpretations of how this model arrives at a particular classification decision. We used the XGBoost library to setup our classification model. In our model, the number of trees was set to 10 and the maximum depth per tree was also set to 10.

### 4.3 Classification metrics

We used three metrics to evaluate the accuracy of classifications. *Precision* is defined as

$$Precision = \frac{t_p}{t_p + f_p}$$

*Recall* is defined as

$$Recall = \frac{t_p}{t_p + f_n}$$

And *f1-score* is defined as the harmonic mean of precision and recall:

$$f1 - score = \frac{t_p}{t_p + \left(\frac{f_p + f_n}{2}\right)}$$

Here $t_p$ is the number of true positives, $f_p$ is the number of false positives, and $f_n$ is the number of false negatives.

### 4.4 Limitations and scope

The dimensionality reduction methods are (1) data-driven and (2) unsupervised. Consequently, there is a very good chance that it will be able to deal with imaging artifacts (presence of clouds, low-light conditions, atmospheric noise, etc.). The classification method is supervised and requires a set of labelled pixels for training. This suggests that if a classifier that is trained using artifact-free pixels is used as is to analyze pixels with artifacts, the performance of this classifier may suffer. On the other hand, it is perhaps possible to improve the performance of this classifier by training it on pixels exhibiting artifacts. We plan to investigate the interplay of hyperspectral pixel compression and classification on data collected under different environmental and lightning setting in the future.

## 5 Experiments and results

A standard way to study the performance of different compression algorithms is to recover the original signal from its compressed version as depicted in Fig 4. In the following sections, we examine reconstruction errors for PCA, KPCA, ICA, AE and DAE for different compression rates. We also present reconstruction errors both with and without the SG noise reduction pre-processing step. As stated earlier, compressing hyperspectral data is desirable; however, we are also interested in pixel-level classification using the compressed data. We define pixel-based classification as the problem of identifying landcover type, say forest, rooftop, etc., for a given pixel in an hyperspectral image. Within this context, we seek the answer to the following two questions: a) how compression rates affect pixel classification scores and b) for a given compression rate, which compression method achieves the highest classification accuracy.

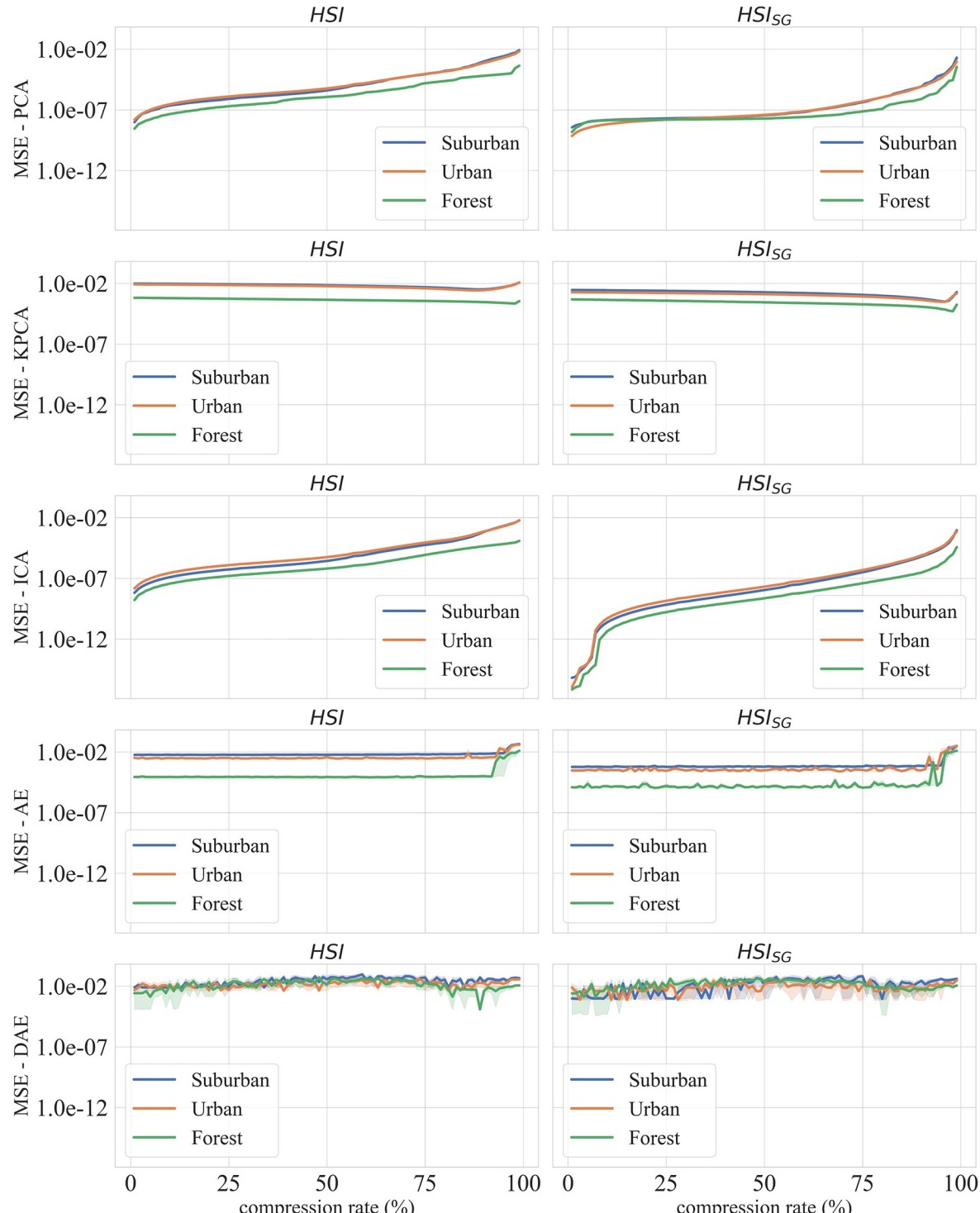

**Fig 4.** MSE (Mean Squared Error) for the 3 datasets and 5 compression algorithms, from top to bottom: a) Principal Component Analysis, b) Independent Component Analysis, c) Kernel Principal Component Analysis, d) AutoEncoder, e) Denoising AutoEncoder.

## 5.1 Spectral reconstruction

Fig 4 presents reconstruction losses for different methods for the three datasets. Here compression rates vary from 1% to 99%. For our purposes, the compression rate is defined as the ratio

of $(n − d)$ to $n$, where $n$ is the number of dimensions of the original signal and $d$ is the number of dimensions of the compressed signal. Recall that $n$ is equal to 301 for the datasets used in this paper. Compression rates are related to the memory needed to store the compressed data. The left column of Fig 4 shows the results for the original hyperspectral data (HSI), whereas the right column shows the results for the data that have been pre-processed using the SG filter (HSI+SG).

The reconstruction errors rise as compression rates increase for both PCA and ICA models. Notice, however, other methods—KPCA, AE, and DAE—are able to achieve low reconstruction errors even for high compression rates. This effect can be explained by the fact that non-linear methods are able to better handle non-linearities present in the data while PCA and ICA are linear methods. It is interesting to note that PCA, KPCA, and ICA methods outperform deep learning methods AE and DAE for compression rates less than ninety percent. Furthermore, AE and DAE match the reconstruction performance of PCA, KPCA, and ICA only for compression rates higher than ninety percent.

The difference between reconstruction errors for original data (HSI) and for data preprocessed using SG filter (HSI+SG) falls as compression rate increases. This is noteworthy since it suggests that compression may have a denoising effect on the original spectral signal. Curiously, we also observe a slightly higher variance in reconstruction score for DAE method for pre-processed data (HSI+SG), which merits further investigation, and we leave it as future work. In the following section, we do not apply SG to the original spectra before compression and classification, as it does not show effective as discussed in the Section 4.1.

## 5.2 Classification

The reconstruction error is a measure of how much information is preserved in the compressed signal, since this information is needed to reconstruct the original signal. The reconstruction error, however, is not a robust measure of classification performance on the compressed signal. In this section we study classification performance on compressed signal for PCA, KPCA, ICA, AE and DAE compression methods and for different compression rates.

Fig 5 shows the confusion matrices for landcover classification for RGB data. Here bands corresponding to red ($670nm$), green ($540nm$), and blue ($470nm$) wavelengths are selected to form RGB pixels. These scores provide a baseline for the classification results obtained by using the hyperspectral data. Fig 6 shows *f1-scores*, precision, and recall values obtained using 1) RGB, 2) uncompressed, and 3) compressed hyperspectral data. The results shown for compressed data are aggregated over all compression rates. RGB *f1-scores* range between 0.9 and 0.96; however, *f1-scores* obtained by using hyperspectral data fall between 0.96 and 0.98. Fig 7 shows *f1-scores*, and it confirms our intuition that the classification results obtained by using hyperspectral data are better than those obtained by using RGB channels. We will return to this later in this section.

We now turn our attention to the case when classification is performed on compressed hyperspectral data. Fig 8 plots *f1-scores* for PCA, KPCA, ICA, AE and DAE compression methods vs. compression rates. In each case an XGBoost classifier is used to predict pixel land-cover-types. A total of 1470 classifiers is trained, one for each compression rate (98) for every compression method (5) and for each dataset (3), in order to ensure that the differences in classification scores can be explained by the ability of the compression algorithm to encode the relevant information. Each XGBoost classifier is trained using identical meta parameters and training regimes. These experiments then provide a different lens for studying compression algorithms. Specifically, these experiments help us pose the question: is it true that

(A) Suburban, RGB

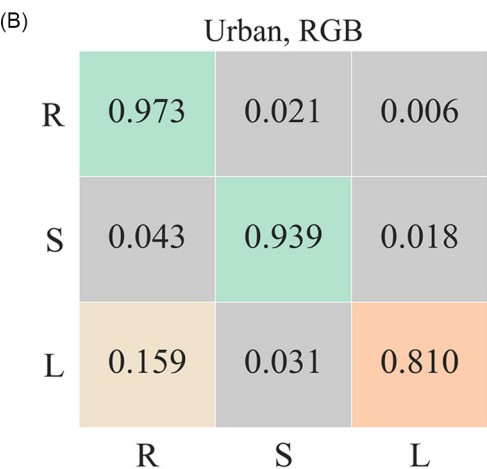

(B) Urban, RGB

(C) Forest, RGB

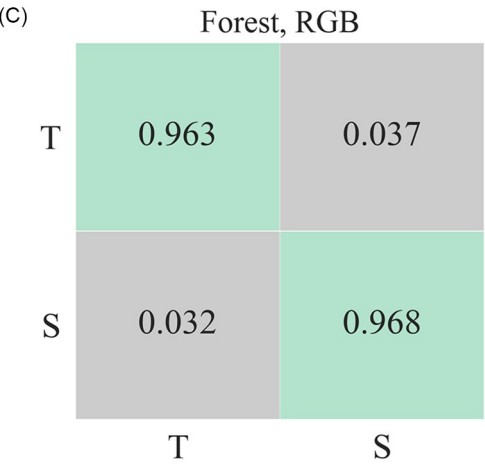

**Fig 5. Confusion matrix for classification scores for three datasets using RGB data.** (Left) R, V, S, and A refer to Rooftop, Vegetation, Shadow, and Asphalt; (Center) R, S, L refer to Rooftop, Shadow and Lawn, respectively; and (Right) T, S refer to Tree and Shadow, respectively.

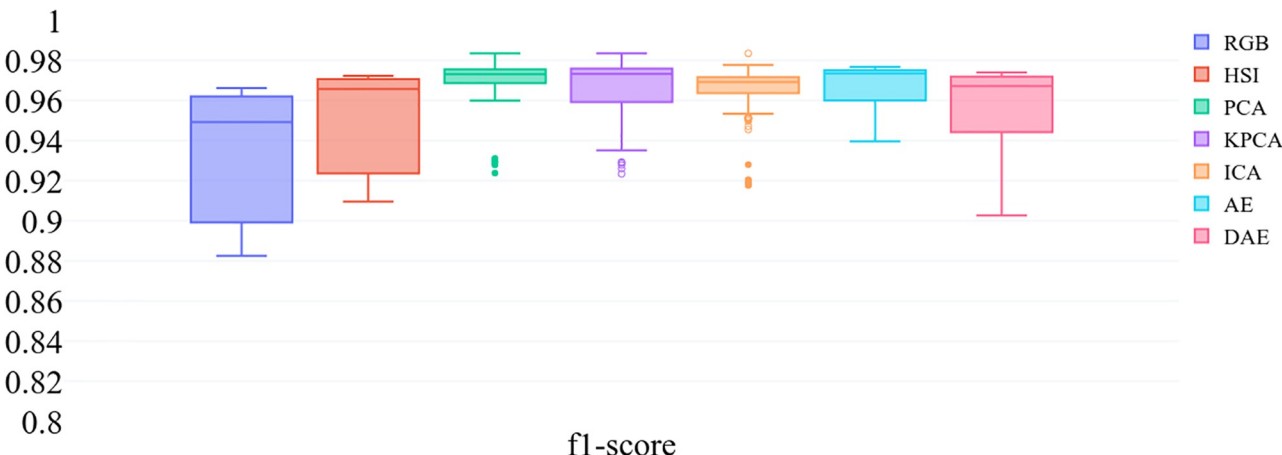

**Fig 6. F1-scores across all compression rates for all datasets and landcover types using PCA, KPCA, ICA AE and DAE methods.** This figure also includes precision, recall and f1 score for all datasets and landcover types when using RGB data for pixel classification.

compression algorithms that achieve low reconstruction errors also create a compressed signal that encodes the information necessary to perform pixel-level classifications?

Ideally, we want classification algorithms that operate in the compressed signal domain. It is both computation and space inefficient to have to reconstruct the original signal to perform classification. As expected, classification performance as measured by *f1-scores* drops as compression rates increase. At the same time, however, nearly all methods post *f1-scores* greater than 0.85 even for compression rates greater than eighty percent. This suggests that it is possible to achieve good classification performance when using a compressed hyperspectral signal.

**5.2.1 Classification using RGB data.** The RGB classifier only achieves an accuracy of 69% for rooftop landcover type in the suburban dataset. Using hyperspectral data improves upon the classification scores obtained by using RGB data. These results confirm that *f1-scores* for hyperspectral datasets improve upon those for RGB data by around one to two percent. Note also that this improvement is maintained when performing classification using the compressed data. Specifically, our results suggest that this improvement holds even at 98% compression rates. At 98% compress rate, each hyperspectral pixel is encoded in a 6-dimensional vector, which is only twice the number that is needed to store an RGB pixel. We believe that classification scores using hyperspectral data, compressed or otherwise, will pull ahead of the scores obtained by RGB data as the number of landcover types (or labels) increases. We currently do not have access to a dataset that is needed to study this issue further.

## 5.3 Classification on compressed data (Suburban dataset)

Fig 9 visualizes classification *f1-score*, recall, and precision values for compression rates between 1% and 99% on Suburban dataset. As expected scores for PCA, KPCA, AE, and DAE compression methods decrease as the compression rate increases. ICA is an outlier. ICA has higher classification performance for compression rates between 63% and 77%. PCA compression achieves best classification performance for compression rates less than 90%. AE compression method achieves best classification performance when compression rate is between 95% and 97%. While classification performance of AE and DAE compression methods is similar to other methods for low compression rates, AE and DAE achieve better classification

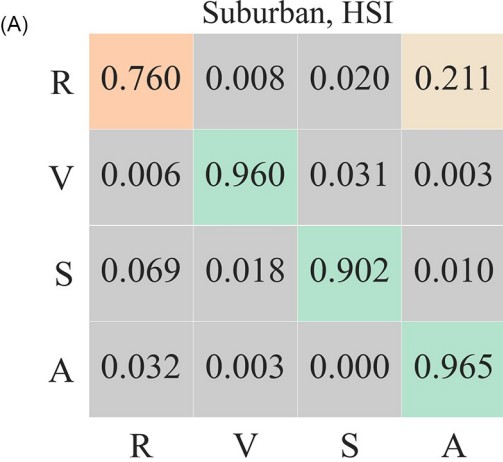

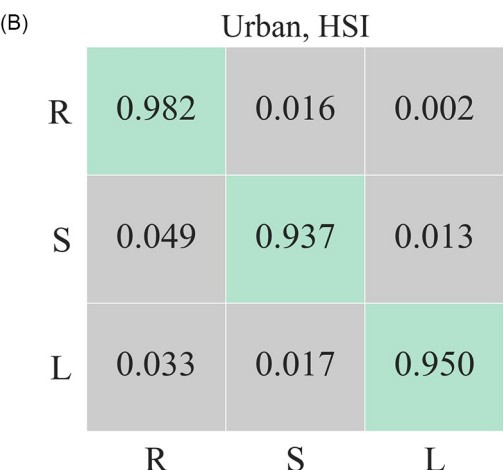

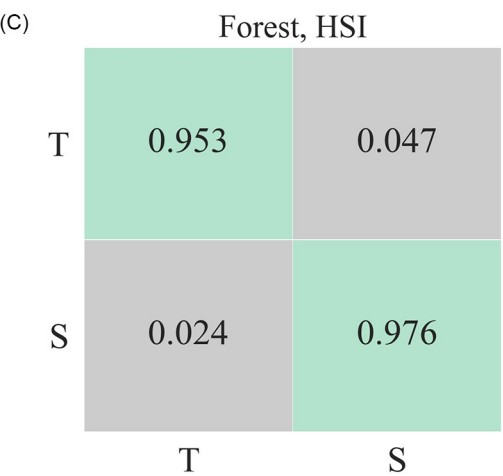

**Fig 7. Confusion matrix for classification scores for three datasets using HSI data.** (Left) R, V, S, and A refer to Rooftop, Vegetation, Shadow, and Asphalt; (Center) R, S, L refer to Rooftop, Shadow and Lawn respectively; and (Right) T, S refer to Tree and Shadow.

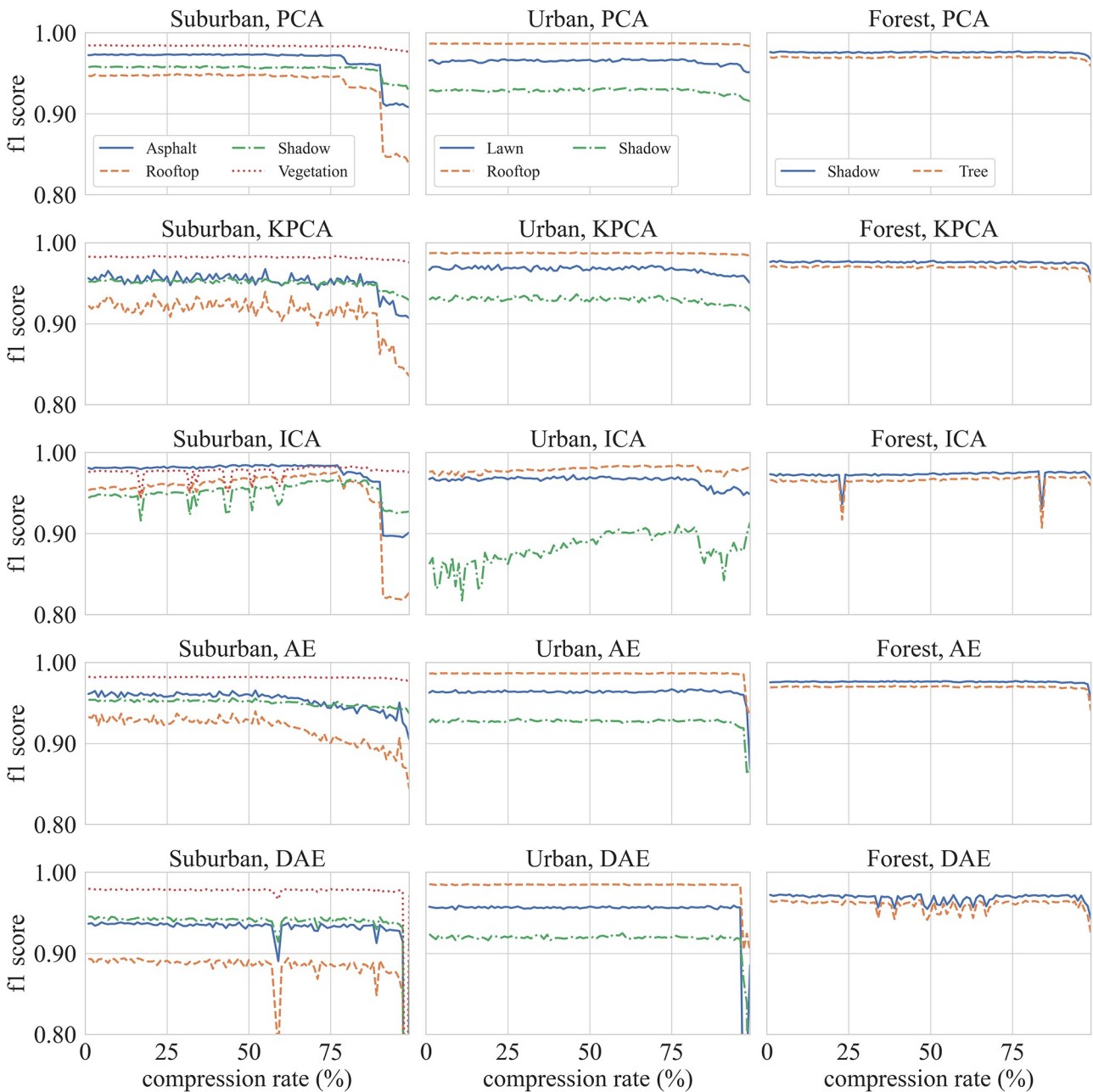

**Fig 8. Classification f1 scores (using compressed data) vs. compression rates.** First column plot results for the Suburban dataset, the second column plot results for the Urban dataset, and the last column plot results for the Forest dataset. The rows group the classification algorithms. The *f1-scores* are plotted for each label present in the dataset.

performance as compared to those obtained by other methods for compression rates between 90% to 95%. Classification accuracy plummets for compression rates greater than 97%. Table 4 shows *f1-scores*, precision, and recall for Suburban dataset at 95% compression (the compressed signal is a 15-dimensional vector, down from 301-dimensional original spectral signal). It also includes these scores for the RGB data. AE and DAE methods outperform other methods at this compression rate.

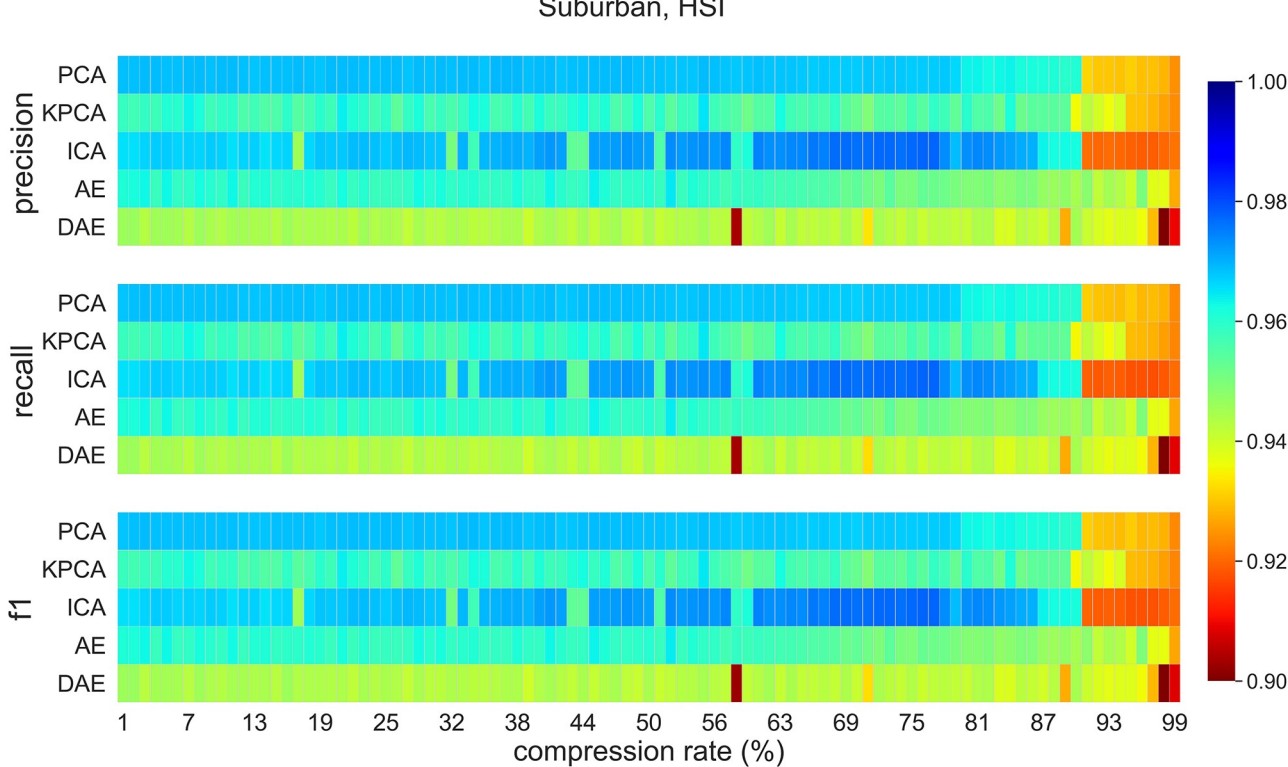

**Fig 9. Suburban dataset classification scores for all methods for compression rates between 1% to 99%.**

### 5.4 Classification on compressed data (Urban dataset)

Fig 10 visualizes classification *f1-score*, recall, and precision values for compression rates between 1% and 99% on Urban dataset. PCA and KPCA are the best performing methods; however, AE is able to match the performance of these methods for compression rates less than 96%. ICA method seems to be struggling with this dataset. The performance of the DAE method is inconsistent across the compression rates range. This can be attributed to the stochastic nature of this method. The classification accuracy for data compressed using ICA is higher for compression rates between 63% and 77%. For compression rates of less than 90%, best classification scores are achieved when data are compressed using PCA. AE compression method achieves the best classification scores when the compression rate lies between 95% and 97%. Table 5 shows *f1-score*, precision, and recall for the urban data at the 95% compression. The table also includes these scores for the RGB baseline. AE and DAE methods outperform other methods at this level of compression.

### 5.5 Classification on compressed data (Forest dataset)

Fig 11 visualizes classification *f1-score*, recall, and precision values for compression rates between 1% and 99% on Forest dataset. DAE and ICA perform poorly on this dataset. PCA, KPCA, and AE compression methods achieve good classification performance on this dataset, where AE outperforming PCA and KPCA methods at 97% compression. Classification performance for data compressed using ICA method curiously improves with compression rate. This merits further investigation. Table 6 shows *f1-score*, recall, and precision for the Forest

**Table 4. Top classification scores Suburban, HSI, compression rate = 95%.**

| label | compression | precision | recall | f1-score |
|---|---|---|---|---|
| **Asphalt** | RGB | 0.809 | 0.968 | 0.881 |
| | PCA | 0.888 | 0.939 | 0.913 |
| | KPCA | 0.885 | 0.939 | 0.911 |
| | ICA | 0.862 | 0.934 | 0.897 |
| | AE | 0.917 | 0.950 | 0.928 |
| | DAE | 0.914 | 0.943 | 0.928 |
| **Rooftop** | RGB | 0.776 | 0.693 | 0.732 |
| | PCA | 0.856 | 0.845 | 0.851 |
| | KPCA | 0.855 | 0.842 | 0.849 |
| | ICA | 0.823 | 0.816 | 0.819 |
| | AE | 0.881 | 0.887 | 0.878 |
| | DAE | 0.873 | 0.880 | 0.877 |
| **Shadow** | RGB | 0.952 | 0.881 | 0.915 |
| | PCA | 0.955 | 0.918 | 0.936 |
| | KPCA | 0.954 | 0.917 | 0.935 |
| | ICA | 0.956 | 0.896 | 0.925 |
| | AE | 0.960 | 0.928 | 0.943 |
| | DAE | 1.000 | 0.923 | 0.938 |
| **Vegetation** | RGB | 0.945 | 0.933 | 0.939 |
| | PCA | 0.980 | 0.978 | 0.979 |
| | KPCA | 0.979 | 0.977 | 0.978 |
| | ICA | 0.978 | 0.975 | 0.977 |
| | AE | 0.981 | 0.980 | 0.980 |
| | DAE | 0.977 | 1.000 | 0.977 |

dataset at the 95% compression. It also includes these values for the RGB baseline. AE and DAE methods outperform other methods at this level of compression.

## 5.6 Computational considerations

The compression methods used in this work need to be trained before these can be used to compress the incoming 301-dimensional spectral signal. The methods only need to be trained once; therefore, the training time is not important as the compression time (the time it takes these methods to encode the incoming signal into a low-dimensional space). Fig 12 shows compression and training times for each method. Notice that KPCA has the largest compression time, which suggests that KPCA is not well-suited to applications where smaller runtimes are desirable.

While the training datasets used in this paper fit in memory, one can imagine a situation where the size of data excludes this possibility. It is not easy to use PCA, KPCA, and ICA in situations where training data do not fit in memory. Deep learning methods, such as AE and DAE, can be trained in batches; therefore, these methods can be trained in situations where the entire training data does not fit in memory. Fig 13 shows that AE and DAE have better scalability properties than other compression methods. This suggests that AE and DAE methods may be more suited for in-situ applications where computational resources are usually limited.

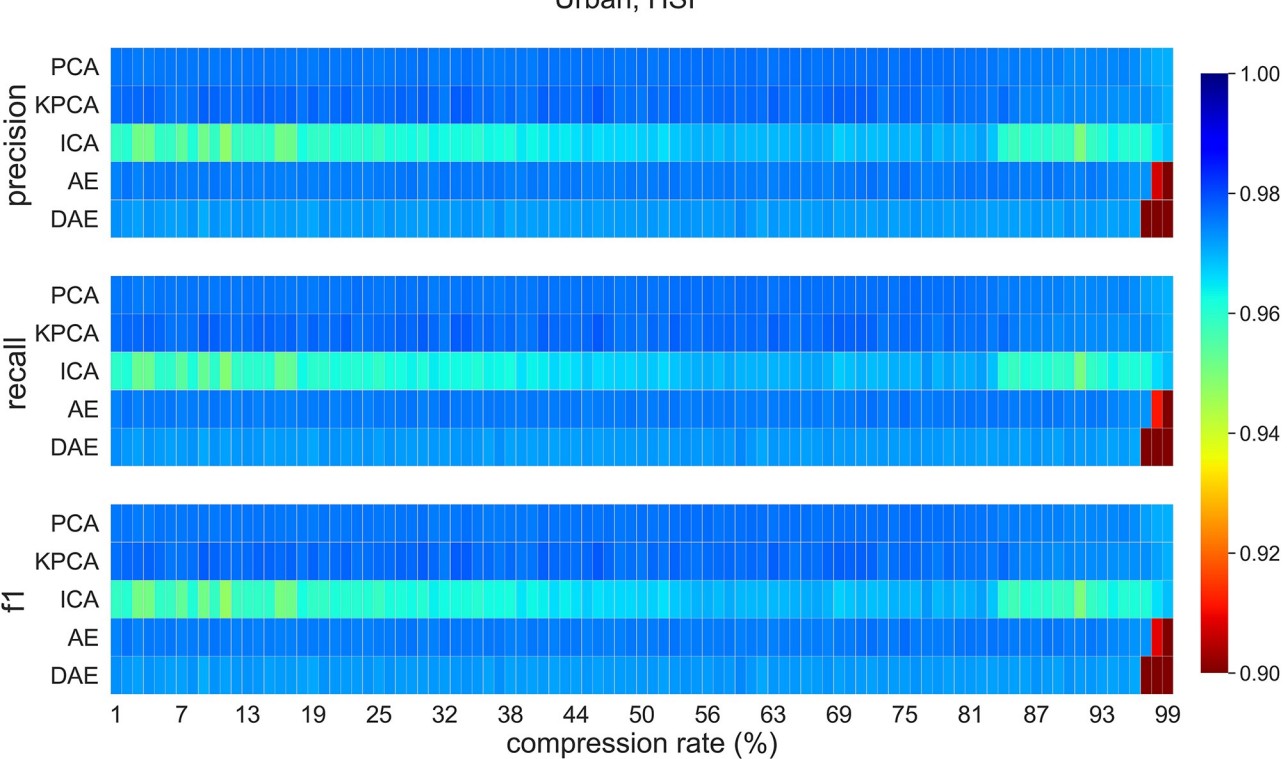

**Fig 10. Urban dataset classification scores for all methods for compression rates between 1% to 99%.**

**Table 5. Top classification scores Urban, HSI, compression rate = 95%.**

| label | compression | precision | recall | f1-score |
|---|---|---|---|---|
| **Lawn** | **RGB** | 0.930 | 0.810 | 0.866 |
| | **PCA** | 0.963 | 0.960 | 0.962 |
| | **KPCA** | 0.966 | 0.951 | 0.958 |
| | **ICA** | 0.965 | 0.947 | 0.956 |
| | **AE** | 0.967 | 0.957 | 0.962 |
| | **DAE** | 0.961 | 0.954 | 0.957 |
| **Rooftop** | **RGB** | 0.967 | 0.973 | 0.970 |
| | **PCA** | 0.984 | 0.988 | 0.986 |
| | **KPCA** | 0.983 | 0.987 | 0.985 |
| | **ICA** | 0.971 | 0.983 | 0.977 |
| | **AE** | 0.984 | 0.988 | 0.986 |
| | **DAE** | 0.983 | 1.000 | 0.985 |
| **Shadow** | **RGB** | 0.877 | 0.939 | 0.907 |
| | **PCA** | 0.935 | 0.915 | 0.924 |
| | **KPCA** | 0.926 | 0.917 | 0.921 |
| | **ICA** | 0.901 | 0.858 | 0.879 |
| | **AE** | 0.934 | 0.916 | 0.922 |
| | **DAE** | 0.935 | 0.918 | 0.919 |

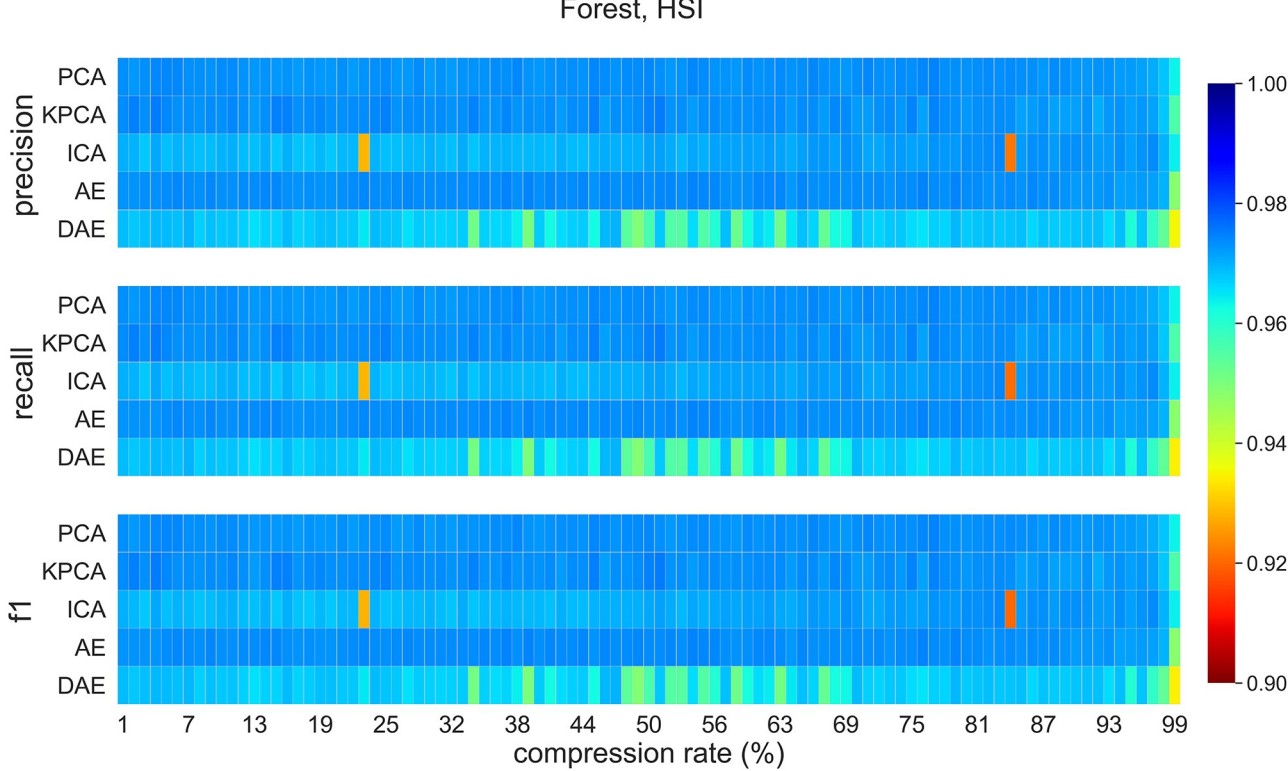

**Fig 11. Forest dataset classification scores for all methods for compression rates between 1% to 99%.**

## 6 Conclusion

Hyperspectral pixels contain two orders of magnitude more information than ordinary RGB pixels, and it is often possible to carry out analysis tasks, such as segmentation and classification, without using the complete spectral signal. As a result, dimensionality reduction techniques, such as PCA, KPCA, ICA, AE, and DAE, are widely employed as a first step in the overall hyperspectral image analysis pipeline. This paper presents a systematic study that

**Table 6. Top classification scores Forest, HSI, compression rate = 95%.**

| label | compression | precision | recall | f1-score |
|---|---|---|---|---|
| **Shadow** | **RGB** | 0.971 | 0.968 | 0.970 |
| | **PCA** | 0.972 | 0.978 | 0.975 |
| | **KPCA** | 0.973 | 0.977 | 0.975 |
| | **ICA** | 0.970 | 0.979 | 0.975 |
| | **AE** | 0.972 | 0.979 | 0.975 |
| | **DAE** | 0.989 | 1.000 | 0.965 |
| **Tree** | **RGB** | 0.960 | 0.963 | 0.962 |
| | **PCA** | 0.973 | 0.964 | 0.969 |
| | **KPCA** | 0.971 | 0.966 | 0.968 |
| | **ICA** | 0.974 | 0.962 | 0.968 |
| | **AE** | 0.973 | 0.964 | 0.968 |
| | **DAE** | 0.964 | 0.992 | 0.956 |

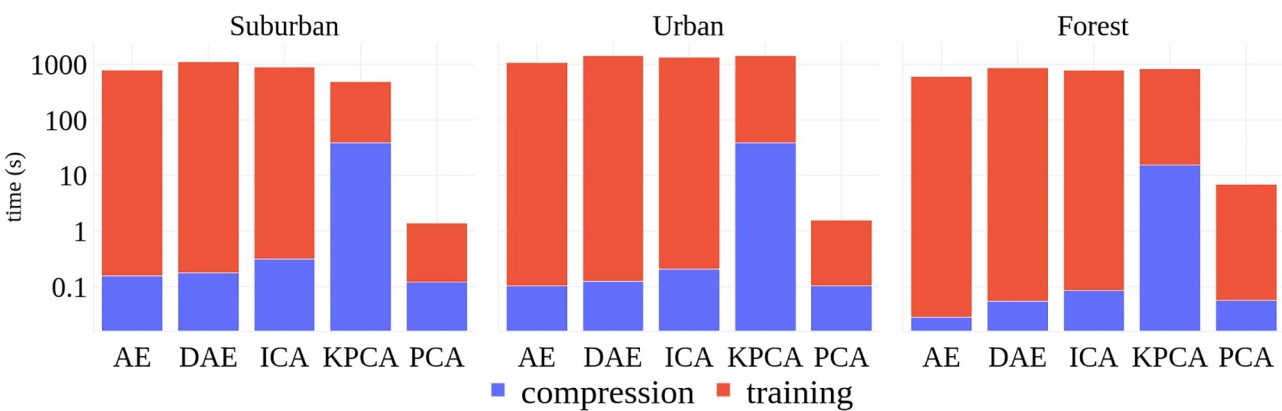

**Fig 12. Execution times (seconds) for training and compression tasks and all compression algorithms.**

investigates the effects of compression on hyperspectral pixel classification. Specifically, we implemented five compression methods—PCA, KPCA, ICA, AE, and DAE—and used these to compress 301-band hyperspectral pixels from three different hyperspectral image datasets. The compression rates varied from 99% to 1%. Gradient Boosted Decision Tree (XGBoost) classifiers were trained for each (compression method, rate, and dataset) yielding a total of 1470 classifiers. Reconstructions scores, classification accuracy, and runtimes for each (compression method, compression rate, classifier, dataset) were recorded to perform an empirical study on the effects of compression on hyperspectral pixel-level classification.

We found that PCA, KPCA, and ICA post lower signal reconstruction errors; however, these methods achieve lower classification scores when the compression rate is greater than 95%. Fig 14 shows the classification results of the three datasets with the compression rates set to 95%. AE and DAE methods post better classification accuracy at compression rates higher than 98%. Noise reduction filtering, which is a common signal preprocessing step for hyperspectral images, is not needed when using DAE for compression. We also captured the runtime performance of different compression methods, and we found that AE and DAE methods are well-suited for resource-constrained, in-situ settings. Our results suggest that the choice of a compression method and compression rate is an important consideration when designing a hyperspectral pixel classification pipeline.

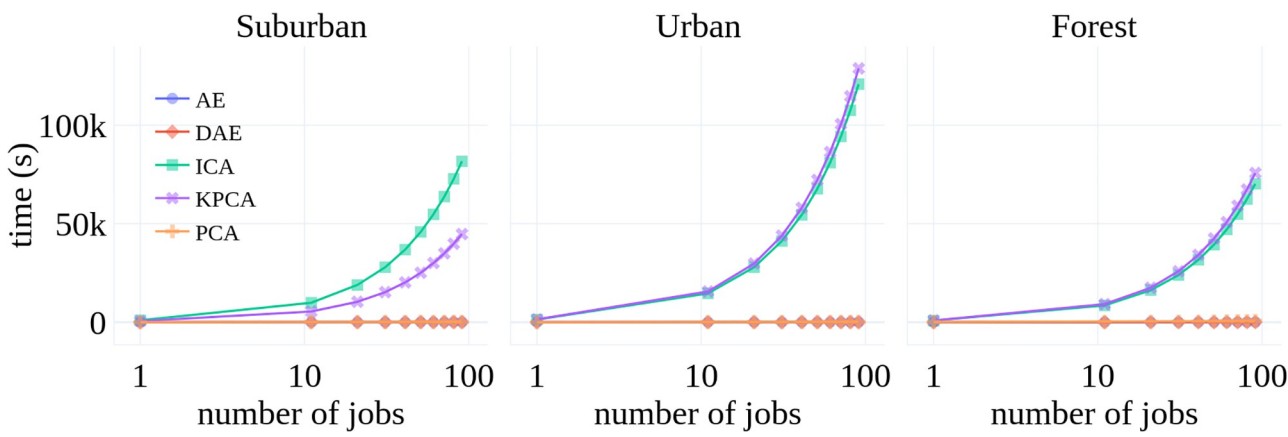

**Fig 13. Execution times (in seconds) times the number of classification jobs.**

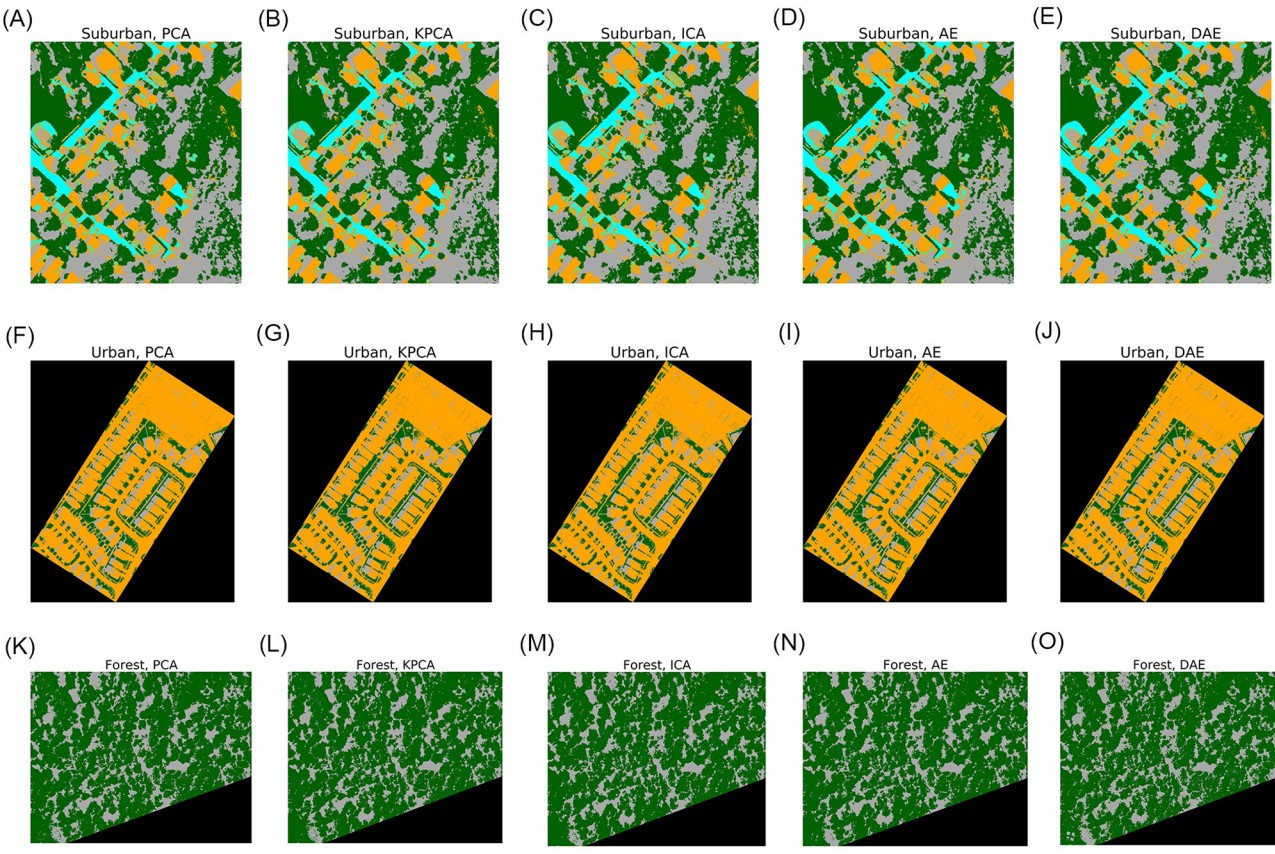

**Fig 14. Classification images with spectra compressed to 95% of original size.**

In the analysis presented in this paper, each hyperspectral pixel is treated independently. In the future, we plan to study Markov Random Field approaches to capture the relationship between neighbouring pixels during hyperspectral pixel classification and image segmentation tasks.

## Author Contributions

**Conceptualization:** Kiran Mantripragada.

**Data curation:** Phuong D. Dao.

**Formal analysis:** Kiran Mantripragada.

**Funding acquisition:** Yuhong He, Faisal Z. Qureshi.

**Investigation:** Kiran Mantripragada.

**Methodology:** Kiran Mantripragada.

**Project administration:** Yuhong He, Faisal Z. Qureshi.

**Resources:** Faisal Z. Qureshi.

**Software:** Kiran Mantripragada.

**Supervision:** Yuhong He, Faisal Z. Qureshi.

**Validation:** Faisal Z. Qureshi.

**Visualization:** Kiran Mantripragada.

**Writing – original draft:** Kiran Mantripragada.

**Writing – review & editing:** Kiran Mantripragada, Phuong D. Dao, Yuhong He, Faisal Z. Qureshi.

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
