## [Decision Letter · Decision Letter 0]

1 Mar 2022

PONE-D-22-02721The Effects of Spectral Dimensionality Reduction on Hyperspectral Pixel Classification: A Case StudyPLOS ONE

Dear Dr. Mantripragada,

Thank you for submitting your manuscript to PLOS ONE. After careful consideration, we feel that it has merit but does not fully meet PLOS ONE’s publication criteria as it currently stands. Therefore, we invite you to submit a revised version of the manuscript that addresses the points raised during the review process. Check the following points:

- The text of the manuscript is intelligible. The English language used must be corrected in several places

- To what extent would the results presented be generalizable to images of the same area made in different seasons or to images obtained in neighboring areas? Are simple correction ratios usable, possibly taking into account weather conditions?

- It would be interesting for authors to make the original images available in a file where interested parties can download and operate them to verify the results.

- The area mapped is represented by a star and does not provide sufficient clarity on the bounds and surface area covered.

We look forward to receiving your revised manuscript.

Kind regards,

Claudionor Ribeiro da Silva

Academic Editor

PLOS ONE

Journal Requirements:

"We acknowledge the support of the Natural Sciences and Engineering Research Councilof Canada (NSERC) through the NSERC Discovery Program for the funding of the

hyperspectral image acquisition mission and image preprocessing facility (RGPIN-386183 awarded to Dr. Yuhong He), and for the Visual Computing Lab of the Ontario Tech University (RGPIN-2020-05159, awarded to Dr. Faisal Z. Qureshi)."

"Natural Sciences and Engineering Research Council of Canada (NSERC) through the NSERC Discovery Program:

Funding of the hyperspectral Image acquisition mission and image preprocessing facility (RGPIN-386183 awarded to Dr. Yuhong He)

- Visual Computing Lab of the Ontario Tech University (RGPIN-2020-05159, awarded to Dr. Faisal Z. Qureshi).

Reviewers' comments:

Reviewer's Responses to Questions

**Comments to the Author**

1. Is the manuscript technically sound, and do the data support the conclusions?

Reviewer #1: Yes

Reviewer #2: Yes

2. Has the statistical analysis been performed appropriately and rigorously? 

Reviewer #1: Yes

Reviewer #2: Yes

3. Have the authors made all data underlying the findings in their manuscript fully available?

Reviewer #1: Yes

Reviewer #2: Yes

4. Is the manuscript presented in an intelligible fashion and written in standard English?

Reviewer #1: Yes

Reviewer #2: Yes

5. Review Comments to the Author

Reviewer #1: The manuscript describes a very laborious scientific research activity, specific to a relatively new field, namely hyperspectral image processing, very voluminous information entities that float in many application fields: geography, geology, topometry, image processing, creating maps with various applications. , imaging medicine, etc.

Processing images from satellite, drone or aeroplane, to RGP images, or something more complicated, requires complex statistical methods (PCA, KPCA, ICA, AE, DAE). Then the images compressed to the small informational size are subjected to the initial image recovery tests, at a loss level as low as possible. To estimate image compression and initial image recovery performance, the authors use domain-specific metrics.

The nature of the research performed is one that is based on the numerical experiment of processing the data obtained by photography (photographic images). Adjusting some processing parameters (compression, size reduction, etc.) has empirical consequences because the effects of choosing these parameters cannot be exactly anticipated. Consequently, the research activity is based on the numerical experiment, the estimates of the performances of the chosen procedures being made in statistical terms. This activity results in useful indications for practitioners working in the field of image processing. On the other hand, this approach still has some inexplicable consequences, which turn into variable new strands of research. It would be interesting for the authors to try to answer the question: to what extent would the results presented be generalizable to images of the same area made in different seasons or to images obtained in neighboring areas? Are simple correction ratios usable, possibly taking into account weather conditions?

The conclusions are well argued by the authors, based on overwhelming numerical results and well illustrated in tables and graphs.

There is no additional information submitted by the authors. The processed images, in the already processed form, are presented in fig. 1. In order to make it easier for amateur readers to verify the processing of the original images, the authors should store the original images in an archive where interested parties can download and operate them. Obviously, this archiving is not mandatory, but it is a gateway to dialogue and improvement.

The paper is stored in the archives: https://arxiv.org/abs/2104.00788 and https://ui.adsabs.harvard.edu/abs/2021arXiv210400788M%2Fabstract%2F/abstract, but we did not find the original photos in the form of files image.

The text of the manuscript is intelligible. The English language used must be corrected in several places (English uk) either using the features of the MS Word editor, or, for example, the Grammarly free program, or another facility. I reported such minor errors in the reviewed manuscript attached to the review: PONE-D-22-02721_reviewer_rec.pdf

Reviewer #2: The paper is well structured and supported by good illustrations/graphs etc.

There is just one minor suggestion, that the area mapped is represented by a star and does not provide sufficient clarity on the bounds and surface area covered. If this can be changed for better readability and understanding.

6. PLOS authors have the option to publish the peer review history of their article (what does this mean?). If published, this will include your full peer review and any attached files.

Reviewer #1: **Yes: **Petru Cardei

Reviewer #2: No

---

## [Author Response · Author response to Decision Letter 0]

17 Apr 2022

All comments are addressed in the document "Response to Reviewers.pdf"

---

## [Decision Letter · Decision Letter 1]

16 May 2022

The Effects of Spectral Dimensionality Reduction on Hyperspectral Pixel Classification: A Case Study

PONE-D-22-02721R1

Dear Dr. Mantripragada,

We’re pleased to inform you that your manuscript has been judged scientifically suitable for publication and will be formally accepted for publication once it meets all outstanding technical requirements.

Kind regards,

Claudionor Ribeiro da Silva

Academic Editor

PLOS ONE

Additional Editor Comments (optional):

Reviewers' comments:

Reviewer's Responses to Questions

**Comments to the Author**

1. If the authors have adequately addressed your comments raised in a previous round of review and you feel that this manuscript is now acceptable for publication, you may indicate that here to bypass the “Comments to the Author” section, enter your conflict of interest statement in the “Confidential to Editor” section, and submit your "Accept" recommendation.

Reviewer #1: All comments have been addressed

Reviewer #2: All comments have been addressed

2. Is the manuscript technically sound, and do the data support the conclusions?

Reviewer #1: Yes

Reviewer #2: Yes

3. Has the statistical analysis been performed appropriately and rigorously? 

Reviewer #1: (No Response)

Reviewer #2: Yes

4. Have the authors made all data underlying the findings in their manuscript fully available?

Reviewer #1: Yes

Reviewer #2: No

5. Is the manuscript presented in an intelligible fashion and written in standard English?

Reviewer #1: Yes

Reviewer #2: Yes

6. Review Comments to the Author

Reviewer #1: I think the last two remaining parentheses can be removed, but if the authors think they are useful they can leave them as they are. Mathematically, they have no effect.

Otherwise, everything is OK, it can be published.

Reviewer #2: My comments have been addressed. And the paper seems to have improved from its original version. The maps and illustrations are excellent.

7. PLOS authors have the option to publish the peer review history of their article (what does this mean?). If published, this will include your full peer review and any attached files.

Reviewer #1: **Yes: **Petru Cardei

Reviewer #2: **Yes: **Salman Atif

---

## [Editor Report · Acceptance letter]

1 Jul 2022

PONE-D-22-02721R1 

The Effects of Spectral Dimensionality Reduction on Hyperspectral Pixel Classification: A Case Study 

Dear Dr. Mantripragada:

I'm pleased to inform you that your manuscript has been deemed suitable for publication in PLOS ONE. Congratulations! Your manuscript is now with our production department. 

Kind regards, 

on behalf of

Dr. Claudionor Ribeiro da Silva 

Academic Editor

PLOS ONE